# The Impact of Cytokines on the Health-Related Quality of Life in Patients with Systemic Lupus Erythematosus

**DOI:** 10.3390/jcm8060857

**Published:** 2019-06-15

**Authors:** Warren David Raymond, Gro Østli Eilertsen, Sharanyaa Shanmugakumar, Johannes Cornelis Nossent

**Affiliations:** 1Rheumatology Group, School of Medicine, The University of Western Australia, Perth 6009, Australia; sharonshanmugakumar@gmail.com (S.S.); johannes.nossent@uwa.edu.au (J.C.N.); 2Molecular Inflammation Research Group, Department of Clinical Medicine, Arctic University, 9037 Tromsø, Norway; gro.eilertsen@uit.no; 3Department of Rheumatology, Sir Charles Gairdner Hospital, Perth 6009, Australia

**Keywords:** SLE, quality of life, patient assessment of global disease activity, cytokines

## Abstract

Introduction: Systemic lupus erythematosus (SLE) reduces the health-related quality of life (HRQoL), even during periods of disease quiescence. We investigated whether subclinical inflammation as reflected by cytokine levels is linked with reduced HRQoL. Methods: A cross-sectional study of SLE patients (*n* = 52, mean age 47.3, 86.5% female) who completed a Short Form Health Survey-36 (SF-36) questionnaire. The clinical and demographic data, scores for the disease activity (SLEDAI-2K), organ damage (SDI), and laboratory data were collected simultaneously. The autoantibody and cytokine levels (IFN-γ, IL-1β, IL-4, IL-6, IL-10, IL-12, IL-17, BAFF, TNF-α, TGF-β1, MIP-1α, MIP-1β and MCP-1 (levels in pg/mL) were quantified by sandwich ELISA. The comparisons and associations were assessed non-parametrically, and a multiple regression determined the effect sizes (ES) of the variables on the SF-36 domain and summary scores. Results: The SF-36 summary and domain scores for SLE patients were significantly (20–40%) lower than in a comparable control group, with the exception of the Mental Health scores (*p* = 0.06). SLE patients had a normal body mass index (BMI) (median, 24.2 kg/m^2^), a high rate of smoking (69.2%), and usage of social security benefits (90.4%). TGF-β1 (ES 0.06), IL-12 (ES −0.11), IFN-γ (ES 0.07) and MCP-1 (ES 0.06) influenced the SF-36 domain scores; and MCP-1 (ES 0.04) influenced the Mental Health Summary Score (MCS). Obvious manifestations, including patient visual analogue scale (VAS) (ES −2.84 to −6.29), alopecia (ES −14.89), malar rash (ES −14.26), and analgesic requirement (ES −19.38), independently influenced the SF-36 items; however, the SF-36 scores were not reflected by the physician VAS or disease activity (SLEDAI-2K). Conclusions: Cytokines had a minimal impact on HRQoL in SLE patients, especially compared to visible skin manifestations, central nervous system (CNS) damage, and pain. Better tools are needed to capture HRQoL in measures of disease activity.

## 1. Introduction

Systemic lupus erythematosus (SLE) is a systemic autoimmune condition with an unclear aetiology, with wide ranging manifestations. SLE leads to a poorer health-related quality of life (HRQoL) by negatively affecting the patient’s physical and mental health [1]. The reduction of HRQoL experienced by SLE patients is equivalent to patients with other chronic diseases, such as acquired immunodeficiency and congestive heart failure [2].

Patients with SLE experience a reduction of HRQoL early in their disease, which continues for at least 5 years following diagnosis [3]. The initial reduction in HRQoL has been associated with a range of disease-specific complications and psychosocial factors [2,3,4,5,6,7,8,9,10,11]. Importantly, the HRQoL reduction in SLE patients persists in the absence of an active disease, the need of glucocorticoids, or accrued damage [12], which indicates that clinically quiescent or well-controlled SLE can still reduce HRQoL [13]. The reduction of HRQoL in the absence of obvious physical manifestations may result from the persistent immune dysfunction in SLE; which contributes to the upregulation of proinflammatory cytokines, such as interleukin 6 (IL-6) and tumour necrosis factor α (TNF-α), which have been associated with fatigue, irritability, depressive moods and social withdrawal [14]. Given that the current disease activity tools are not well suited to capture the underlying immunological activity [15], this study investigated the influence of cytokines on HRQoL using the SF-36 domain and summary scores in patients with SLE.

## 2. Methods

This cross-sectional study of patients in the Tromso Lupus Cohort (*n* = 52, mean age 47.3, 86.5% female) who completed a Short Form Health Survey-36 (SF-36) questionnaire during an outpatient visit. SF-36 has been validated for use in Norwegian studies [16] and is comprised of 8 dimension scales: physical function (PF), role limitations due to physical problems (role physical, RP), bodily pain (BP), general health (GH), vitality (VT), social function (SF), role limitations due to emotional problems (role emotional, RE), and mental health (MH) [17]. Each scale ranges from 0 (lowest) to 100 (highest). These 8 dimensions can be summarised into two global scores, the physical component summary (PCS) and the mental component summary (MCS). The scores for patients in our study were compared to Norwegian reference values for women (*n* ≥ 1097) in 2015 [18].

All study patients fulfilled the American College of Rheumatology’s (ACR) classification criteria for SLE [19,20]. During the same visit as the SF-36 reporting, data was also collected on clinical characteristics, the disease activity with the Systemic Lupus Erythematosus Disease Activity Index—2000 (SLEDAI-2K) [15], and accrued organ damage with the Systemic Lupus International Collaborating Clinics—Damage Index (SDI) [21]. The disease activity was also scored on a 0–10 cm visual analogue scale (VAS) scale by the patient (patVAS) and attending physicians (pVAS).

The disease activity was grouped as remission (SLEDAI-2K = 0), low disease activity (SLEDAI-2K 1 ≤ x ≤ 4), and high disease activity (SLEDAI-2K > 4). Organ damage was grouped as no damage (SDI = 0), moderate damage (SDI 1 ≤ x ≤ 3), and severe organ damage (SDI > 3). Medication use was grouped as: glucocorticosteroids (GC), anti-malarials (AM), and immunosuppressives (IS), which included: azathioprine, mycophenolate, methotrexate, rituximab or calcineurin inhibitors. We defined, “government benefits” as receiving a disability pension or a stipend for vocational training; “cardiovascular events” were defined as the occurrence of a heart attack or blood clots; “gastrointestinal problems” included the requirement of proton pump inhibitors, irritable bowel or diarrhoea; “mental health problems” as the presence or need for the treatment for depression, insomnia, anxiety or other mood disorders; and “omega-3 supplementation” as any use of seal, fish or cod liver oil.

Serum samples were collected at a routine blood monitoring and stored in 1 ml aliquots at −70 degrees, and they underwent 1–2 freeze-thaw cycles only. Anti-double-stranded DNA antibodies (anti-dsDNA Ab) and other autoantibody assays were performed at the clinical immunology laboratory with a validated ELISA (EliATM and VarelisA^®^; Phadia GmbH, Freiburg, Germany). The measurements of interleukin 17A (IL-17A), B-cell-activating factor (BAFF), interleukin 1 beta (IL-1β), interleukin 4 (IL-4), IL-6, interleukin 10 (IL-10), interleukin 12 (IL-12), IFN-γ, macrophage inflammatory protein 1-alpha (MIP-1α), macrophage inflammatory protein 1-beta (MIP-1β), monocyte chemoattractant protein 1 (MCP-1), TNF-α, and transforming growth factor beta 1 (TGF-β1) were done with a quantitative sandwich immunoassay (Single Analyte ELISArray™ kit; SuperArray Bioscience Corp., Frederick, MD, USA). The manufacturer’s recommendations were followed throughout; the same lot was used for each cytokine, and the results were the averaged duplicate assay runs. For statistical purposes, the values below the limit of detection (LOD) were replaced by the LOD value.

### Statistical Analysis

Continuous data are described as either a mean with a standard deviation or a median with an inter-quartile range (IQR); and, categorical data are described as a frequency and proportion. The differences between the groups were assessed with the non-parametric Mann–Whitney U or the Chi-square tests, where appropriate. Spearman’s rank correlation coefficient (Rs) describes the association between the clinical parameters and the SF-36 domain and summary scores. Univariate and multiple regression models determined the effect size (ES) of the clinical variables on the SF-36 domain and summary scores, i.e., the amount change in the SF-36 when the clinical variable increases by 1 unit [22]. Multiple regression models of the SF-36 domains and component summary scores were adjusted for age and disease duration [3,7,23]. A statistical analysis was conducted with IBM SPSS for Windows Version 24.0 (IBM Corporation: Armonk, NY, USA); and, *p*-values (*p*) < 0.05 were considered statistically significant.

## 3. Results

This SLE cohort had a female preponderance (86.5%), a median age of 46.8 years (IQR 32.4, 60.2), and a median disease duration of 10 years ((IQR 5.3, 18.3) range: 0.0, 34.7) (Table 1). Thirty-six patients (69%) smoked a median of 8 cigarettes per day (IQR 5, 10). The patients were mostly high school graduates and averaged 12 years of schooling, yet the majority (90.4%) were reliant on government benefits. The prevalent comorbidities were hypertension (38.5%), gastrointestinal problems (25%), cardiovascular events (34.5%), and psychiatric issues (19%). Patients required prednisone (54%) at a median dose of 6.3 mg/day (IQR 5, 10), antimalarials (69%) and immunosuppressants (60%); and they required anticoagulants (52%) and antihypertensives (46%) to treat the most common comorbidities. The routine blood and urine biomarkers were generally unremarkable (Table 1), with thrombocytopenia present in 2%, leukocytopenia present in 8%, and proteinuria in 15%, respectively. Anti-dsDNA Ab were seen in 39%, and 23% had hypocomplementemia.

The median SLEDAI-2K score was 7 points (IQR 4, 10), while the median patient VAS was 3 points (IQR 2, 5), and the median physician VAS was 2 points (IQR 1, 4). The most frequent clinical symptoms of SLE were malar rash (31%), alopecia (31%), arthritis (25%), and oropharyngeal ulcers (21%). Organ damage (SDI > 0) had occurred in 63.5% (*n* = 33), mostly in the musculoskeletal (27%), cardiovascular (21%), and central nervous (neuropsychiatric) (19%) systems.

SLE patients scored significantly lower on all of the SF-36 domain and summary scores than a Norwegian female reference sample in 2015 [18], but this failed to reach a statistical significance for Mental Health (MH) (Table 2). SLE patients with SLEDAI ≤ 4 (*n* = 18) or SLEDAI > 4 (*n* = 34), as well as SLEDAI ≤ 6 or SLEDAI > 6, were equivalent across all of the SF-36 summary and domain scores (all, *p* > 0.05); however, the SF levels (50 vs. 75, *p* = 0.031) were lower in those with SLEDAI > 10 (*n* = 18) compared to SLEDAI ≤ 10 (*n* = 34). The associations between the SF-36 scores and clinical and biochemical findings are shown in Appendix A. There was a clear negative effect of age and SDI > 0 on PF while the use of osteoporosis medication and analgesics had a large impact on the overall physical health. Age and the use and dosage of prednisone were negatively correlated with mental health. The patVAS was stronger and more broadly correlated with the physical and mental domain scores than pVAS, and SLEDAI-2K was inversely associated with SF only.

### 3.1. Cytokines

The levels of certain cytokines correlated with some SF-36 sub-scales and summary scores but lacked a discernible pattern (Table 3). The PCS summary score was associated with IFN-γ, IL-1β, IL-12, MCP-1, and TNF-α, while the physical domain scores was associated with IFN-γ, IL-1β, IL-12. IL-4, MCP-1, MIP-1β, and TNF-α. The MCS summary score was associated with IL-12, MCP-1 and MIP-1β, while the mental domains were associated with IL-1β, IL-12, MCP-1 and MIP-1β. In the univariate linear regression models, PF increased with increasing levels of IFN-γ and IL-12. RP increased with increasing levels of TGF-β1. BP, GH or the PCS were unaffected by changing cytokine levels. VT increased with increasing levels of MCP-1. RE increased with increasing levels of BAFF and MIP-1β. MH increased with increasing levels of MCP-1. The MCS increased with increasing levels of MCP-1 and MIP-1β. However, all statistically significant effect sizes were quite small in absolute terms.

### 3.2. Multiple Regression Model

After adjusting for age and disease duration, the PCS worsened with an increasing patient VAS (ES –4.09), analgesic use (ES –19.38), alopecia (ES –14.89) and platelet count (ES –0.09) (Table 4). The physical domains were reduced by a worsening patVAS, the presence of alopecia, increased steroid requirement, IS use, analgesic use, and falling levels of IFN-γ and TGF-β1. Similarly, MCS worsened with an increasing patient VAS (ES –2.84) or malar rash (ES –14.86), but improved slightly with MCP-1 levels (ES 0.04). The mental domains were reduced by a worsening patVAS, IS use, malar rash, damage accrual, falling MCP-1 level, falling C3 level, greater daily cigarette consumption, and increasing HbA1c (%). 

## 4. Discussion

Patients in the Tromso Lupus Cohort demonstrated significantly lower SF-36 scores than a normative Norwegian female population. The reduction in HRQoL was best captured by the patient’s own assessment of the global disease activity (patVAS), to a lesser extent by the physician VAS score, and, marginally by the total SLEDAI-2K score, which was only associated with the SF domain. While reduced TGF-β1, IFN-γ and MCP-1 levels and increased IL-12 levels were significantly associated with lower HRQoL scores, there were much larger absolute independent effect sizes on age adjusted HRQoL for the presence of malar rash or alopecia, analgesic use, daily cigarette consumption, and poor glycaemic control (HbA1c (%)), compared to the unitary changes in the cytokine levels.

Relative to the normative female Norwegian population, and in line with the literature, SLE patients had a 20–40% reduction in all of the SF-36 domains, with the exception of MH [7,12,23,24,25,26]. The PCS and MCS scores were similar to other SLE cohorts, with the exception of a higher MCS compared to the PCS scores [23,24,27,28]. This could be due to the low RP scores, which to date are the lowest reported in SLE, and correlated with the PCS, MCS and each domain (all, Rs > 0.35), with the exception of RE and MH [12,28]. The dissociation of the mental HRQoL items herein might reflect the socioeconomic and healthcare factors in Norway, coupled with the malleability of mental health over time. This is especially relevant given that 90.4% of participants utilised social security services, which in spite of our insignificant findings, were shown to be protective of HRQoL [1,23]. However, given our higher average RE compared to RP scores, we suspect that greater societal acceptance and reduced stigmatism toward those who could benefit from social security services, especially those with chronic diseases like SLE, may have masked the true effects of these programmes on HRQoL [6,27,29]. Second, the Norwegian healthcare system offered free psychiatric outpatient care to help patients accept the potential for lifestyle changes caused by a chronic disease. Finally, mental HRQoL may have been protected through the regular clinical appointments over an average follow-up time of 13 years, which would have provided patients with a disease-specific stress reduction and coping strategies [12,13,26,30,31].

Our patients had a comparable age, gender, and disease duration to other studies reporting on HRQoL in SLE [23,24,28]. We confirmed the negative association of age and disease duration within the physical HRQoL items [3,6,7,23,24,28]. With the high rates of smoking compared to other studies, we confirmed that an increasing daily cigarette consumption leads to worsened MH [7,13,32]. However, whether smoking contributes to or is a reflection of worse MH remains an open question. Omega-3 supplementation (40.4%) did not influence HRQoL, which was confirmatory given the data showing a lack of efficacy of omega-3 supplementation in patients with SLE [33].

The cytokine levels were equivalent across patients with different levels of disease activity and organ damage. This contradicted other studies, but this may have been due to the small sample size, as well as to different definitions of the outcome measures [4,13]. The weak but positive correlations between cytokines and HRQoL were mostly attenuated in the regression model, with the exception of TGF-β1, IFN-γ, IL-12, and MCP-1. Increasing TGF-β1 levels were associated with higher RP in SLE. While this is a novel finding in the context of HRQoL, it aligns with the evidence demonstrating that SLE patients produce less TGF-β1 than healthy individuals [34], and that a deficiency in TGF-β1 can contribute to the onset and exacerbation of SLE [34,35]; thus, pathways to optimise TGF-β1 may help improve HRQoL in SLE. IFN-γ levels have been associated with disease flares and lupus nephritis, while we found that the IFN-γ level was mildly associated with physical functioning [35,36,37]. Anti-IFN-γ therapy was inefficacious in SLE and resulted in more adverse events and fatigue; however, the HRQoL data was not formally captured [38]; in spite of this, oncology patients receiving anti-IFN-γ therapy had a similar increase of adverse events and demonstrated worsened HRQoL [38,39,40]. Increasing IL-12 levels were seen with a greater SLE disease activity [41] and were associated with worsened HRQoL in rheumatoid and psoriatic arthritis [42,43]; thus, its association with decreased PF in SLE, is a novel, yet unsurprising finding. Increasing MCP-1 increased VT and MCS, which contradicts the data on the disease activity in SLE, where the serum, cerebrospinal and urinary MCP-1 levels were increased with disease exacerbation; and MCP-1 was also shown to be associated with fatigue and depression in SLE; while simultaneously beingnegatively associated with all SF-36 items, (all, *p* < 0.01) [4]. Our results were similar to other studies which showed that the MCP-1 levels remain elevated after a long-term immunosuppression, even when the disease activity had subsided [44,45,46]. More generally, MCP-1 levels have been shown to be equivalent between women with and without Exhaustion Disorders, anxiety or depression [47]; and MCP-1 is important for combating bacterial infections [48]. Thus, the beneficial effect of MCP-1 on HRQoL seen here may represent a protection against infection in the context of chronic inflammation and IS requirement. However, compared to the clinical parameters, the absolute effect size of cytokine levels on HRQoL was very low, suggesting a large overlap of cytokine level between various HRQoL states, and indicates that a substantial change in cytokine level would be required to cause meaningful differences to a patient’s HRQoL.

Given the limited association for cytokines, we performed ad-hoc analyses of the clinical, serological, immunological data to determine their influence on HRQoL in patients with SLE.

While the rates of comorbidities, such as Raynaud’s phenomenon, cardiovascular events, hypertension, gastrointestinal problems, thyroid dysfunction, and mental health problems were comparable to other studies, in contrast to other studies they did not influence HRQoL [2,13,26,49,50,51,52,53,54,55].

Medication usage was overall comparable to other studies [7,56,57]; and while confirming that antimalarial usage had no effect on HRQoL, prednisone requirement and dosage was associated with lower PF, as shown by Chaigne et al. [7], and IS requirement negatively impacted GH, which confirmed earlier findings [57]. Analgesic requirement was negatively associated with BP and PCS, which confirmed that pain or pain syndromes, such as fibromyalgia, can reduce HRQoL in SLE [7,26,58]. We found no impact of anticoagulation [55], antihypertensives (no comparable data), mood stabilisers [55], and NSAIDs [59] on HRQoL. For the routine laboratory findings, we found that HbA1c (%) negatively impacted MH, aligning with the glycaemic control literature [60,61], but that the levels of proteinuria and anti-dsDNA antibodies were not related to HRQoL [7,26,62,63]. An increasing C3 level led to higher SF; however, the levels or hypocomplementemia of C3 or C4 did not influence any other HRQoL item. Those with low C3 (*n* = 11) had higher SLEDAI-2K (8 vs. 6, *p* = 0.021), clinical SLEDAI-2K (6 vs. 4, *p* = 0.046; which excludes hypocomplementemia and the presence of anti-dsDNA Abs), and more oropharyngeal ulcers (63.6% vs. 9.8%, *p* < 0.01). Given that oropharyngeal ulcers are not associated with low C3 [61]; and, we demonstrated that SF was reduced by an increasing disease activity (patVAS), especially new malar rash, and was negatively associated with the SLEDAI-2K score (Rs −0.31, *p* = 0.03); this would suggest that significant complement activation of C3, perhaps driven by a fulminant disease activity or potentially by other non-immune mediated pathways, may contribute to reduced SF.

The reported SLEDAI-2K scores and physician VAS scores were comparable to other studies, but were dissociated from HRQoL, as reported elsewhere [24,56,57,62,64]. In contrast, the patVAS scores, which were similar to the global scores in comparable studies [6,24], had the greatest impact on HRQoL. This points towards an important discrepancy between the “objective” and “subjective” scoring of disease activity, whereby the SLEDAI-2K and physician VAS are primarily weighted to the prognostically important, immunological, renal and CNS manifestations, whereas the patVAS better captures the current impact of disease on the health status and functioning. This is nicely illustrated by the negative impact of alopecia and new onset malar rash on HRQoL in this study, which may be less appreciated by physicians, yet the presence of these manifestations would hinder participation in either schooling or the workplace.

Finally, organ damage accrual was similar to other studies [23,24,57] and showed an independent reduction of PF and SF, especially for those who sustained neuropsychiatric damage. The negative impact on PF was also found by Hanly et al. as part of a broader reduction of the mental HRQoL items [24,56,57].

The generalisability of our findings are limited by the cross-sectional study design, which does not allow for the study of longitudinal changes in disease-related and external factors. Furthermore, the relatively small sample size may have a limited statistical power, and we lacked uniform data on fibromyalgia, which has been shown to impact HRQoL in SLE [6,58]. The strength of this study lies in the comprehensive and synchronized data collection in a homogeneous Caucasian cohort a with well-controlled disease.

## 5. Conclusions

HRQoL in Norwegian SLE patients is significantly lower than in the general population; but does not affect mental health to the same degree as seen in other cohorts, possibly as a result of the chronic disease support, including social security in the Norwegian health system The reduced HRQoL was better captured by patient VAS scores than by global physician and SLEDAI-2K scores. Clinical and lifestyle factors have a larger effect on HRQoL than cytokine levels.

## Figures and Tables

**Table 1 jcm-08-00857-t001:** Systemic lupus erythematosus (SLE) patient demographic and clinical characteristics at the time of study.

Demographic and Clinical Characteristics at the Time of Study	SLE Patients (*n* = 52)	
	Median (f)	*n* (%)
Age (years)	46.8 (35.4, 60.2)	
Time since diagnosis (years)	10 (5.3, 18.3)	
Female		45 (86.5)
Schooling (years)	12 (10.0, 15.0)	
Required social security		47 (90.4)
Body mass index (BMI)	24.2 (21.8, 27.8)	
Smoker		36 (69.2)
Number of cigarettes daily	8 (5, 10)	
Systolic BP (mm Hg)	130 (119, 142)	
Diastolic BP (mm Hg)	80 (66, 85)	
Comorbidities		
Cardiovascular events		18 (34.6)
Stroke		6 (11.5)
Hypertension		20 (38.5)
Mental Health Problems		10 (19.2)
Gastrointestinal conditions		13 (25.0)
Thyroid dysfunction		5 (9.6)
Medication		
Prednisone daily		28 (53.4)
Prednisone dose (mg/day)	6.3 (5.0, 10.0)	
Hydroxychloroquine		36 (69.2)
Immunosuppressants		31 (59.6)
Non-steroidal anti-inflammatory drugs (NSAIDs)		11 (21.2)
Anticoagulants		27 (51.9)
Anti-hypertensives		24 (46.2)
Statins		8 (15.4)
Anti-resorptives		9 (17.3)
Mood stabilising drugs		9 (17.3)
Analgesics		7 (13.5)
Fish oil		4 (7.8)
Serology		
Anti-dsDNA positive		20 (39.2)
Hypocomplementemia		12 (23.1)
HbA1c (%)	5.7 (5.5, 6.0)	
Hemoglobin (g/dL)	12.8 (12.0, 13.9)	
White Blood Cells (10^9^/L)	6.2 (4.3, 7.4)	
Lymphocytes (mm^3^)	1 (1.0, 2.0)	
Erythrocyte Sedimentation Rate (mm/h)	18.5 (11.5, 34.0)	
High sensitivity C-Reactive Protein (mg/L)	2.9 (0.7, 8.5)	
Creatinine (mmol/L)	71 (45, 92)	
Systemic Lupus Erythematosus Disease Activity Index—2000 (SLEDAI-2K) score	7 (4, 10)	
Patient visual analogue scale (VAS)	3 (2, 5)	
Physicians VAS	2 (1, 4)	
Systemic Lupus International Collaborating Clinics—Damage Index (SDI) > 0		33 (63.5)
Median SDI	2 (1, 4)	

Required social security = disability pension or vocational training stipend; Cardiovascular events = occurrence of heart attack or blood clots; Gastrointestinal problems = use of proton pump inhibitors, irritable bowel or diarrhoea; Mental health problems = treatment for depression, insomnia or anxiety; Immunosuppressants = azathioprine, mycophenolate, methotrexate, rituximab or calcineurin inhibitors; Hypocomplementemia = C3 < 0.84 g/L or C4 < 0.08 g/L. IQR: inter-quartile range. BP: bodily pain.

**Table 2 jcm-08-00857-t002:** Short Form Health Survey-36 (SF-36) Summary Measures and domain scores for (and within) patients in the Tromso Lupus Cohort against a Norwegian reference value from 2015. Figures indicate mean ± standard deviation.

	Norwegian Female, 2015 (*n* = 1149) (A)	Overall SLE (*n* = 52) (B)	SLEDAI-2K ≤ 4 (*n* = 18) (C)	SLEDAI-2K > 4 (*n* = 34) (D)	Independent *t*-Test A vs. B	Independent *t*-Test C vs. D
Physical Functioning	84.9 ± 21.0	66.35 ± 24.50	72.78 ± 21.09	62.94 ± 25.76	<0.001	0.171
Role-physical	72.6 ± 39.6	27.88 ± 36.60	27.78 ± 40.12	27.94 ± 35.23	<0.001	0.988
Bodily Pain	66.9 ± 26.5	53.74 ± 27.38	56.44 ± 27.19	52.31 ± 27.79	<0.001	0.609
General Health	72.6 ± 22.5	44.56 ± 23.53	49.72 ± 24.17	41.82 ± 23.07	<0.001	0.253
Vitality	57.2 ± 20.6	38.94 ± 24.80	39.44 ± 25.26	38.68 ± 24.93	<0.001	0.917
Social Functioning	85.7 ± 21.6	62.98 ± 27.11	69.44 ± 26.16	59.56 ± 27.36	<0.001	0.214
Role-emotional	87.4 ± 28.6	66.67 ± 41.22	74.07 ± 42.09	62.75 ± 40.84	<0.001	0.351
Mental Health	79.9 ± 14.8	76.00 ± 15.68	81.33 ± 13.23	73.18 ± 16.32	0.061	0.074
Physical Summary		48.13 ± 21.83	51.68 ± 22.00	46.25 ± 21.83	-	0.399
Mental Summary		61.15 ± 19.39	66.07 ± 19.74	58.54 ± 18.98	-	0.185

**Table 3 jcm-08-00857-t003:** Spearman rank correlation and linear regression coefficients for the overall strength (Rs) and effect size (ES) of the relation between the inflammatory cytokine levels (pg/ml) and health related quality of life, quantified with the Short Form 36 domain and summary scores.

Cytokine		Physical Domains				Mental Health Domains				PCS	MCS
		PF	RP	BP	GH	VT	SF	RE	MH		
BAFF	Rs/ES	−0.17/ NS	−0.24/NS	−0.13/NS	−0.27/NS	0.07/NS	−0.12/NS	0.23/**0.091 ***	0.18/NS	−0.23/NS	0.16/NS
IFN-γ	Rs/ES	**0.31 */0.04 ***	**0.31 ***/NS	0.24/NS	0.17/NS	0.09/NS	0.16/NS	0.09/NS	−0.16/NS	**0.33 ***/NS	0.10/NS
IL-1β	Rs/ES	**0.34 ***/NS	**0.30 ***/NS	0.19/NS	0.13/NS	0.16/NS	0.24/NS	**0.28 ***/NS	−0.05/NS	**0.33 ***/NS	0.27/NS
IL-4	Rs/ES	0.10/NS	0.28 */NS	0.16/NS	0.10/NS	0.22/NS	0.12/NS	0.17/NS	−0.10/NS	0.24/NS	0.20/NS
IL-6	Rs/ES	0.14/NS	0.13/NS	0.18	0.12/NS	0.07/NS	0.15/NS	0.06/NS	−0.05/NS	0.17/NS	0.11/NS
IL-10	Rs/ES	0.02/NS	0.21/NS	−0.01/NS	0.12/NS	0.07/NS	0.11/NS	0.23/NS	0.07/NS	0.13/NS	0.23/NS
IL-12	Rs/ES	**0.30 */0.07 ***	**0.31 ***/NS	0.26/NS	**0.37 ****/NS	0.22/NS	**0.30 ***/NS	0.25/NS	0.12/NS	**0.38 ***/NS	**0.33 ***/NS
MCP-1	Rs/ES	0.15/NS	**0.30 ***/NS	0.25/NS	0.19/NS	**0.34 */0.064 ***	0.19/NS	0.18/NS	0.25/**0.034 ***	**0.32 ***/NS	**0.34 */0.05 ***
MIP-1α	Rs/ES	0.02/NS	0.06/NS	0.20/NS	0.12/NS	−0.05/NS	0.14/NS	−0.07/NS	−0.11/NS	0.13/NS	−0.06/NS
MIP-1β	Rs/ES	0.10/NS	0.06/NS	0.15/NS	0.17/NS	**0.32 ***/NS	0.26/NS	0.25/**0.077 ***	**0.29 ***/NS	0.15/NS	**0.35 */0.040 ***
TNF-α	Rs/ES	0.27/NS	**0.50 ****/NS	0.01/NS	**0.32 ***/NS	0.00/NS	0.15/NS	0.20/NS	−0.07/NS	**0.32 ***/NS	0.16/NS
TGF-β1	Rs/ES	0.05/NS	0.14/**0.04 ***	0.08/NS	0.04/NS	0.07/NS	0.07/NS	0.11/NS	0.20/NS	0.11/NS	0.11/NS

*: *p* < 0.05; **: *p* < 0.01; NS: not statistically significant, *p* ≥ 0.05 PF: Physical Function; RP: Role-Physical; BP: Bodily Pain; GH: General Health; VT: Vitality; SF: Social Functioning; RE: Role-Emotional; MH: Mental Health; PCS: Physical Component Summary; MCS: Mental Component Summary.

**Table 4 jcm-08-00857-t004:** Multiple linear regression model, assessing the effect size (ES) of the significant univariate factors on the health-related quality of life, quantified with the Short Form 36 domain and summary scores.

	Physical Scales				Mental Health Scales				PCS	MCS
	PF	RP	BP	GH	VT	SF	RE	MH		
Age	−0.39 (−0.73, −0.05)	−0.82 (−1.43, −0.20)	NS	NS	NS	NS	NS	NS	NS	NS
Pat. VAS (1–10)	−3.36 (−5.20, −1.53)	NS	−6.29 (−8.86, −3.72)	−4.99 (−7.28, −2.70)	−3.72 (−6.34, −1.09)	−4.83 (−6.96, −2.70)	NS	−3.10 (−5.76, −0.43)	−4.09 (−5.78, −2.40)	−2.84 (−4.79, −0.89)
Number of Cigarettes per day	NS	NS	NS	NS	NS	NS	NS	−1.58 (−2.81, −0.35)	NS	NS
Alopecia	NS	−20.52 (−38.88, −2.15)	NS	NS	NS	NS	NS	NS	−14.89 (−23.35, −6.44)	NS
Malar Rash	NS	NS	NS	NS	NS	−18.95 (−30.20, −7.70)	NS	NS	NS	−14.86 (−24.50, −5.22)
SDI > 0	−13.60 (−23.19, −4.01)	NS	NS	NS	NS	−14.63 (−25.15, −4.12)	NS	NS	NS	NS
NP damage	−15.71 (−27.68, −3.73)	NS	NS	NS	NS	NS	NS	NS	NS	NS
Prednisone. (mg/d)	−1.77 (−2.71, −0.825)	NS	NS	NS	NS	NS	NS	NS	NS	NS
IS drugs	NS	NS	NS	−17.15 (−28.13, −6.16)	NS	NS	NS	NS	NS	NS
Analgesics	NS	NS	−22.67 (−40.70, −4.65)	NS	NS	−22.90 (−38.44, −7.36)	NS	NS	−19.38 (−31.00, −7.76)	NS
HbA1c	NS	NS	NS	NS	NS	NS	NS	−6.28 (−11.37, −1.19)	NS	NS
C3	NS	NS	NS	NS	NS	37.24 (13.22, 61.27)	NS	NS	NS	NS
Thrombocytes	NS	−0.22 (−0.34, −0.10)	NS	NS	−0.10 (−0.18, −0.01)	NS	NS	NS	−0.09 (−0.14, −0.03)	NS
TGF-β1	NS	0.06 (0.02, 0.09)	NS	NS	NS	NS	NS	NS	NS	NS
IFN-γ	0.071 (0.024, 0.118)	NS	NS	NS	NS	NS	NS	NS	NS	NS
IL-12	−0.112 (−0.222, −0.002)	NS	NS	NS	NS	NS	NS	NS	NS	NS
MCP-1	NS	NS	NS	NS	0.06 (0.01, 0.11)	NS	NS	NS	NS	0.04 (0.01, 0.08)

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
