# Peer review of "The Impact of Cytokines on the Health-Related Quality of Life in Patients with Systemic Lupus Erythematosus"

_jcm, 2019, doi:10.3390/jcm8060857_

Reviewer 1 Report

The article by Raymond et al evaluated the association between several biomarkers with quality of life measures (HRQOL) in a group of SLE patients. The article is of interest as it looks at the role subclinical inflammation may play in worsening outcomes in SLE patients and not just overt inflammatory damage/active disease as is often the focus of SLE studies. The methodological approach was also sound, the analysis was appropriate, the discussion was comprehensive including limitations and conclusions were in line with obtained results.

I only have a few minor concerns:

Methods: Authors should clarify how samples were stored prior to teesting. At what temp? For how long? How many freeze-thaw cycles? These variables can affect cytokine levels

Results: Table 1 - Frequency and Median are reported in the same column, which is quite confusing. Please modify table clearly indicating the difference.

Author Response

Thanks for your Review.

Really appreciate the time and effort you spent in helping us improve our manuscript.

We have our responses included in the Cover Letter attached.

We look forward to any further suggestions that you may have to improve the quality of our work.

Kind regards,

Warren

Reviewer 2 Report

Summary

In this manuscript, the authors have modeled health-related quality of life in relation to various factors in context of SLE disease such as cytokines levels in the patients, clinical manifestation and overall physical health. While this has been done before in other cohorts additional data from the Tromso Lupus Cohort will also prove valuable for physicians and researchers in the field. The study will help understand factors that are associate with patient’s personal assessment of their quality of life and also understand that they are different from clinical disease manifestations that do not manifest as obvious perceivable symptoms.

Overall Impression

From a science research point of view the study is well done and well explained. However certain factors may make the article hard to understand for someone outside the field. Certain findings are also unexpected and haven't been explained or discussed by the authors. I have listed these in the point by point concerns. The discussion is very well written.

Point by point concerns

1.     Line 39-40 : “In SLE patients, the reduction of HRQOL occurs within two years from diagnosis, and remain suppressed for at least another 3 years”

It is unclear what “remain suppressed” means. Please rewrite sentence for clarity.

2.     In the discussion please address how BAFF has such large ES when the Rs isn’t statistically significant – how is this possible? What does this imply? Considering that BAFF has the largest ES compared to any other cytokine tested this would be am important point to consider. In this context, also describe the published impact of BAFF in SLE pathogenesis.

3.     Considering C3 has the highest ES, please discuss the immunological context: What role does C3 play in SLE (with references of published studies) and what does this very high effect size to social function imply? What does CS have a large impact on SF and not on any of the physical scales? It is strange that it has no overall impact on MCS in terms of ES when it is so high for SF.

4.     Why is it that as shown in supplementary table 1, mental health issues have no impact on mental scales? That is counter-intuitive. Along the same lines, why does neurological defects have no impact on MCS or PCS? While these two parameters stand out in context of how counter-intuitive they seem, other diagnosed conditions such as Cardiovascular events and Stroke not correlating with the physical or mental scales is also unexpected.

5.     In line 168, “The reduction in HRQOL was captured by the patient’s own assessment of global disease activity (patVAS), but less well by physician VAS scores, and not at all by the SLEDAI-2K.”

It may be incorrect to state “not at all by SLEDAI-2K” because from Supplementary Table 1, SLEDAI-2K is significantly associated with SF which is a mental scale.

Author Response

We have provided a response to all Reviewer Comments in the Cover Letter attached.

Thanks very much for your time and effort in helping us clarify and improve our manuscript.

Really appreciate your help,

Cheers,

Warren
